# A High Sensitivity Electrochemical Immunosensor Based on Monoclonal Antibody Coupled Flower-Shaped Nano-ZnO for Detection of Tenuazonic Acid

Chi Zhang [1], Congcong Du [1], Wei Liu [1], Ting Guo [1], Ying Zhou [1], Hongyuan Zhou [1], Yuhao Zhang [1,2,3], Xiaozhu Liu [4] and Liang Ma [1,2,3,*]

1   College of Food Science, Southwest University, Chongqing 400715, China; zczc1020@email.swu.edu.cn (C.Z.); dcc970205@email.swu.edu.cn (C.D.); lwissue@email.swu.edu.cn (W.L.); guoting06@swu.edu.cn (T.G.); zhouyying@swu.edu.cn (Y.Z.); zhouhy@swu.edu.cn (H.Z.); zhy1203@swu.edu.cn (Y.Z.)
2   Chongqing Key Laboratory of Specially Food Co-Built by Sichuan and Chongqing, Chongqing 400715, China
3   Key Laboratory of Luminescence Analysis and Molecular Sensing (Southwest University), Ministry of Education, Chongqing 400715, China
4   Chongqing Micro Miracles Biotechnology Company, Chongqing 400039, China; qinshuai@tansun.com.cn
*   Correspondence: zhyhml@swu.edu.cn

**Abstract:** In this paper, an electrochemical biosensor was established for the high-sensitivity detection of Tenuazonic acid (TeA) in fruits based on the enrichment of flower-shaped nano-ZnO and the specific recognition of immune response. Herein flower-shaped nano-ZnO (ZnO NFs) with a hexagonal wurtzite structure and diameter of 700–800 nm were demonstrated to have the optimal specific surface area and outstanding conductivity, compared with different morphology, sizes, and crystal structures of nano-ZnO. Second, the ZnO NFs were used as carriers for efficiently immobilizing monoclonal antibodies to obtain antibody bioconjugates, which were anchored on the 2-mercaptobenzoic acid-modified gold electrode by amide reaction. In the presence of TeA, the monoclonal antibody could specifically recognize and bind to it, resulting in a decrease in electron transfer ability on the gold electrode surface. Finally, the electrochemical biosensor showed a range from $5 \times 10^{-5}$ μg/mL to $5 \times 10^{-1}$ μg/mL with a detection limit of $1.14 \times 10^{-5}$ μg/mL. Furthermore, it exhibited high selectivity for TeA among other analogs, such as Altenuene (ALT) and Alternariol (AOH). Notably, the proposed strategy could be employed to monitor TeA in tomato and citrus, showing potential application prospects in practical application and commercial value.

**Keywords:** tenuazonic acid; flower-shaped nano-ZnO; monoclonal antibody; electrochemical immunosensor

## 1. Introduction

Tenuazonic acid (TeA), as one of a variety of secondary metabolites produced by *Alternaria alternate* [1], has potent toxicity compared with the other Alternaria toxins, including cytotoxicity and potential carcinogenicity [2,3]. Moreover, it can produce synergistic toxicity with other mycotoxins, resulting in acute toxicity [4,5]. Actually, TeA widely exists in cereals, pepper, potatoes, grapes, and even animal products, with a pollution level ranging from dozens of micrograms to hundreds of milligrams [6]. Due to its high pollution of agricultural products, TeA was listed in the <Registry of Toxic Effects of Chemical Substance> by the U.S. National Organization for Occupational Safety and Health in 1979. In recent years, it has been reported that the European Union has begun to study and formulate relevant standards [7,8]. In addition, China issued the detection standard of Streptomyces toxin in fruits and vegetables in May 2015 [9]. Up to date, the detection of mycotoxin TeA mainly relied on enzyme-linked immunoassay (ELISA) [10,11] and liquid chromatography–tandem mass spectrometry (LC-MS/MS) [12]. Therefore, improvements

are still required because these classical detection methods retain some disadvantages, including being cumbersome, time-consuming, and having poor sensitivity.

Compared with the above methods, the electrochemical biosensor has been widely identified as a powerful analysis tool with many merits, such as fast detection speed and portable instruments [13–15]. More recently, studies have focused attention on improving its high sensitivity based on the nanomaterial with excellent specific surface area [16], outstanding conductivity [17], and good electrocatalytic properties [18]. Zinc oxide (ZnO), as a universal semiconductor material, possesses dual performance of the nanomaterial and semiconductor material [19–21]. It can significantly provide more reaction sites for biological recognition elements, such as nucleic acid aptamers and monoclonal antibodies, owing to the large specific surface area, strong stability, and good biocompatibility [22,23]. Since different morphologies of ZnO have successfully been applied to the electrochemical biosensor to detect various biomolecules [24–26], it is necessary to compare the effect of different morphologies of ZnO on specific surface areas and conductivity to broaden its application in sensitive biosensors.

Herein, ZnO with different morphologies, sizes, and crystal structures was synthesized in detail by controlling pH, reaction time, reaction temperature, and precipitator. Among them, the flower-shaped nanometer ZnO (ZnO NFs) shows the best load capacity due to the rough surface, three-dimensional arrangement, and many voids on each monomer surface. Subsequently, Scheme 1 shows that the ZnO NFs were employed to construct an electrochemical biosensor for sensitive detection of TeA based on a specific immune response, in which the TeA monoclonal antibody with suitable titer was successfully prepared by immunizing mice with hybridoma cells in a previous study. Specifically, the ZnO NFs were covalently coupled with TeA monoclonal antibody and then immobilized on a gold electrode surface modified with 2-mercaptobenzoic acid (MBA) via an amidation reaction between the amino-group of antibodies and the carboxyl group of MBA. This work first developed an electrochemical sensor to detect TeA. As a result, this developed assay for TeA detection demonstrated excellent sensitivity with a linear range from $5 \times 10^{-5}$ µg/mL to $5 \times 10^{-1}$ µg/mL and limit of detection down to $1.14 \times 10^{-5}$ µg/mL, which provided a method for the rapid on-site detection of TeA.

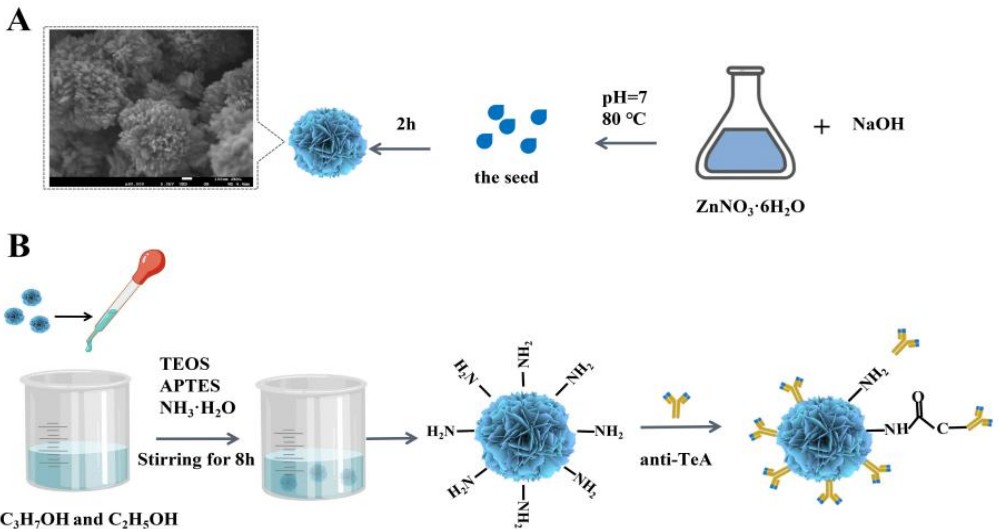

**Scheme 1.** *Cont.*

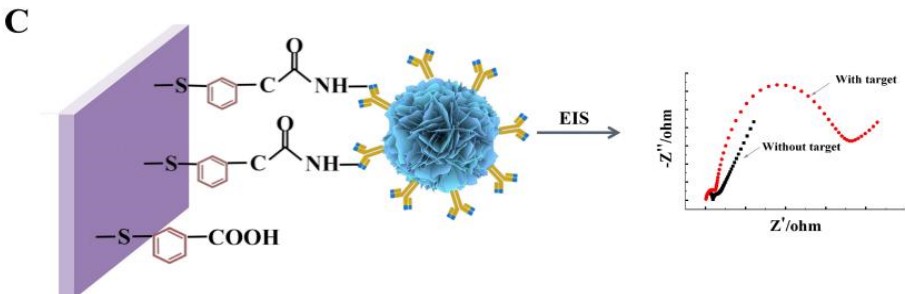

**Scheme 1.** Schematic illustration of the electrochemical biosensor fabrication for detecting TeA. Controllable synthesis of flower-shaped nano-ZnO (**A**); Process of enriching TeA monoclonal antibody with modified flower-shaped nano-ZnO (**B**); EIS characterization of electrochemical sensor performance (**C**).

## 2. Materials and Methods

### 2.1. Materials and Reagents

TeA monoclonal antibody (laboratory patent 2017100291621), zinc nitrate hexahydrate, 1-ethyl-3-(3-dimethylaminopropyl) carbodiimide hydrochloride (EDC), N-hydroxysuccinimide (NHS) (Shanghai Aladdin Biochemical Technology Co., Ltd., Shanghai, China), 2-mercaptobenzoic acid (MBA) (Shanghai Yien Chemical Technology Co., Ltd., Shanghai, China), Bovine albumin (BSA) (Shanghai McLin Biochemical Technology Co., Ltd., Shanghai, China). The butyleyanoacrylate (BCA) protein concentration assay kit was purchased from Shanghai Biyuntian Biotechnology Co., Ltd. (Shanghai, China). The reagents used were analytically pure. The working buffer was 0.1 M phosphate-buffered solution (PBS, containing 10 mM KCl and 2 mM $MgCl_2$, pH 8.0).

### 2.2. Instrument

The transmission electron microscopic (TEM) image was observed on a SU8010 high-resolution field emission scanning electron microscope (Hitachi Ltd., Shanghai, China). Infrared spectroscopy was observed with a Spectrun100 Fourier Transform Infrared Spectrometer (PerkinElmer). The X-ray diffraction (XRD) image was observed on an X 'PERT PRO MPD X-ray diffractometer (Panaco Ltd., Beijing, China). The electrochemical measurements were performed on a CHI 660 E electrochemical workstation (Shanghai, China) at room temperature with a conventional three-electrode system composed of a platinum wire as the counter, Ag/AgCl as the reference, and the gold electrode as working electrode, respectively.

### 2.3. Controllable Synthesis of Nano-Zno

Nano-ZnO was successfully synthesized by hydrothermal reaction using $Zn(NO_3)_2 \cdot 6H_2O$ as the source of zinc, and NaOH and $NH_3 \cdot H_2O$ as the sources of alkali. By changing the conditions, such as pH, temperature, reaction time, and type of precipitant, the relationship between the reaction conditions and the morphology of nano-ZnO was explored, and the controlled synthesis of nano-ZnO was achieved.

### 2.4. Modification of Nano-Zno

Twenty-milligram nano-ZnO was dispersed in a mixture of 20 mL n-propanol and 40 mL ethanol and sonicated for 10 min. Under stirring, 1.5 mL ammonia (25 wt%), 320 μL tetraethyl orthosilicate (TEOS), 80 μL 3-aminopropyl triethoxysilane (APTES) were sequentially added, and the reaction was stirred at room temperature for 8 h. The above solution was centrifuged at 8000 r/min for 10 min to obtain a precipitate. Then the solution precipitate was washed with ultrapure water and dispersed in a PBS solution. The modified material was characterized by infrared spectroscopy.

### 2.5. Loading Capacity of Nano-Zno on Antibody

Point four molar EDC and 0.1 M NHS were added to the modified nano-ZnO suspension, then the anti-TeA monoclonal solution was added to the mixture. The above solution was stirred in a 37 °C water bath for 2 h and centrifuged to obtain a nano-ZnO-monoclonal antibody (ZnO-mAb). The BCA protein concentration determination kit was used to detect the protein concentration in the supernatant, and the amount of antibody loaded on the nano-ZnO was calculated by the Lambert–Beer law (1) formula and the subtraction method [27].

$$\text{Absorbance (A)} = a\,L\,c \tag{1}$$

A is the absorbance coefficient, L/(gm), L is the distance the light travels in the sample (usually the thickness of the cuvette), cm, c is the concentration of the solution, g/L

### 2.6. Construction of Tea Electrochemical Immunosensor

A conventional three-electrode cell setup was used in which a gold electrode was the working electrode, Ag/AgCl was the reference electrode, and a Pt wire served as the counter electrode. Cyclic Voltammetry (CV) scanning was used at a scanning speed of 50 mV/s. Five millimolar $K_3[Fe(CN)_6]/K_4[Fe(CN)_6]$ (1:1)solution containing 0.1 M KCl was used as supporting electrolyte.

First, the gold electrode was immersed in 1% MBA ethanol solution, incubated at 37 °C for 2 h, rinsed with ultrapure water, and dried with nitrogen. Then, the gold electrode was immersed in 0.4 M EDC and 0.1 M NHS to incubate at 37 °C for 0.5 h, rinsed with ultrapure water, and dried with nitrogen. After that, 5 µL of the ZnO-mAb solution was dropped on the gold electrode to incubate at 37 °C for 2.5 h, then rinsed with ultrapure water, and dried with nitrogen. Next, 5 µL of 5% BSA PBS solution was dropped onto the gold electrode to incubate at 37 °C for 1.5 h, rinsed with ultrapure water, and dried with nitrogen. After the modification of each layer above was completed, a CV scan and Electrochemical impedance spectroscopy (EIS) were also carried out with the CHI660E electrochemical analyzer. The scan frequency range was set to 0.01–100 kHz. Finally, 5 µL of TeA acetonitrile solution was dropped onto the gold electrode to incubate at 37 °C for 1 h, rinsed with ultrapure water, and dried with nitrogen. Differential pulse voltammetry (DPV) investigated the electrochemical performance with a step potential of 4 mV, a frequency of 25 Hz, and an amplitude of 25 mV by scanning the potential from 0 to 0.8 V.

### 2.7. Actual Sample Detection

Tomato and oranges were selected as actual samples for the spike recovery experiment. Pretreatment of actual samples refers to relevant literature with a slight improvement [28,29]. A one-gram sample was added to different concentration gradients of TeA standard solution in addition to the blank group, and then 1 mg solid pectinase was added to enzymatic hydrolysis for 3 h at 50 °C. After that, 5 mL of 1.5% formic acid-acetonitrile solution was added and ultrasonicated for 30 min at 20 °C. In the end, 0.5 g anhydrous $MgSO_4$ and 0.1 g NaCl were added, shaken for 1 min, and centrifuged at 1000 r/min. After 5 min, the supernatant was passed through a 0.22 µm organic filter membrane and then tested.

## 3. Results and Discussion

### 3.1. Characterization of Nano-Zno

SEM was carried out to investigate the morphology of nano-ZnO. The results showed that the reaction pH and reaction temperature had significant effects on the morphology of nano-ZnO. With an increase in the alkalinity of the reaction system, the morphology of ZnO gradually changed from particles to rods. When the temperature reached 80 °C, Figure 1A shows that ZnO looked like a "flower" because it grew from particles to a sheet with the increase in temperature. When the pH reached 13, Figure 1B shows that ZnO grew in numerous clusters, and the surface began to become rough, forming a dense lamellar structure in the same direction, which was very similar to the "brush". Among them,

brush-shaped and flower-shaped nano-ZnO had great application potential because they could load more biometric elements in the detection with the advantage of a large specific surface area.

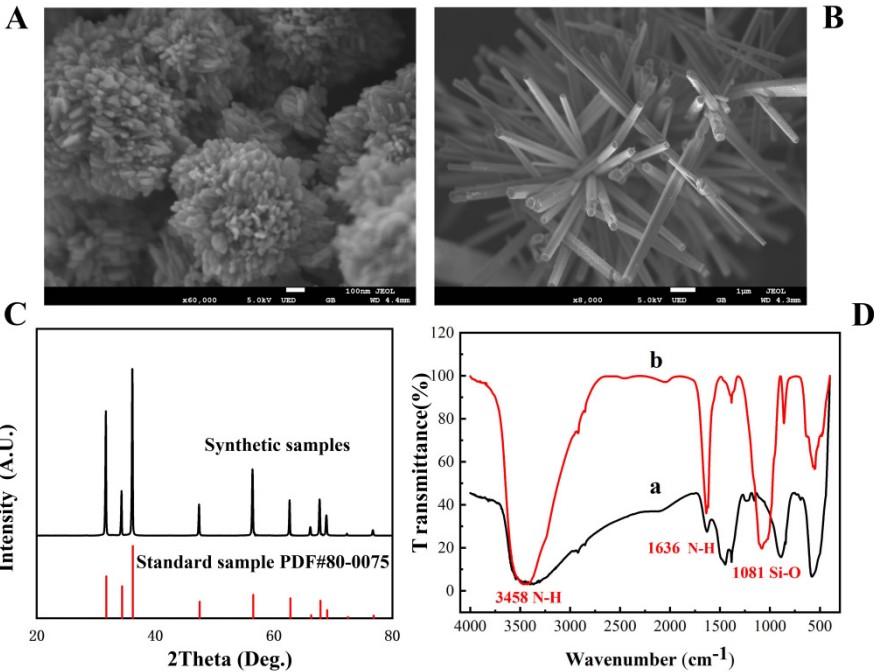

**Figure 1.** The SEM image of flower-shaped nano-ZnO (**A**) and brush-shaped nano-ZnO (**B**); XRD patterns of ZnO (**C**); FT-IR spectrum of amino-modified nano-ZnO (**D**).

The structure of nano-ZnO was measured by XRD, and the result is shown in Figure 1C; the XRD pattern of nano-ZnO was consistent with the standard card PDF#80–0075 [30], indicating that the structure of ZnO was a hexagonal wurtzite structure.

The amino modification of nano-ZnO was divided into two steps. First, tetraethyl orthosilicate was hydrolyzed to form $SiO_2$ coated on the surface of ZnO, and then amino was provided by 3-aminopropyltriethoxysilane to modify the nano-ZnO. The success of the modification can be judged by the characteristic peaks of the corresponding groups in the infrared spectrum. Figure 1D(a,b) show the infrared spectra of the nano-ZnO material before and after amino modification, respectively. As shown in Figure 1D(b), the asymmetric stretching vibration peak of Si-O-Si at 1081 cm$^{-1}$ was observed, indicating that $SiO_2$ had been coated on the surface of ZnO. The peak at 3458 cm$^{-1}$ and 1636 cm$^{-1}$ were observed, which were associated with stretching vibration and flexural vibration peak of N-H, indicating that the amino group had been successfully modified on the surface of the flower-shaped nano-ZnO material.

### 3.2. Comparison of Loading Capacity of Nano-Zno on Antibody

The number of antibodies loaded on ZnO was limited. We investigated the loading capacity of different forms of nano-ZnO on TeA monoclonal antibodies. As shown in Figure 2, we can see that the number of monoclonal antibodies on the surface of flower-shaped nano-ZnO increased with the increase in the input ratio. When the input ratio was greater than 14:1, the loading ratio tended to be stable, illustrating that each milligram of flower-shaped nano-ZnO can load up to 6.63 mg of TeA monoclonal antibody. Similarly, the loading amount of brush-shaped nano-ZnO reached a saturated state when the mass ratio was 10:1, illustrating that each milligram of brush-shaped nano-ZnO can load up to 3.33 mg of TeA monoclonal antibody. It was found that the loading capacity of flower-shaped nano-ZnO was better than that of brush-shaped nano-ZnO for TeA monoclonal antibodies.

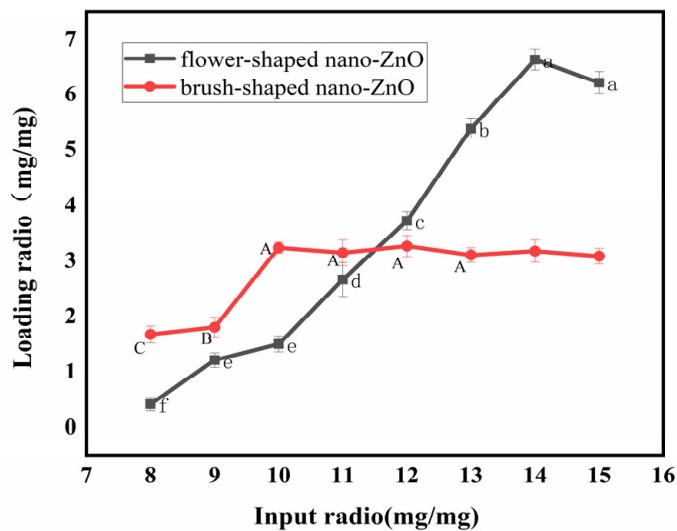

**Figure 2.** The loading capacity of TeA monoclonal antibodies on nano-ZnO with different morphology.

### 3.3. Construction of the Electrochemical Immunosensor

CV and EIS were used to monitor the electrochemical immunosensor fabrication (Figure 3A). Compared with the bare gold electrode (curve a), the peak current decreased after modifying with MBA on the gold electrode, indicating a block of charge transfer due to the formed organic layer. After ZnO-mAb was immobilized on the surface of the electrode, the peak current continued to decrease (curve c). Because nano-ZnO had poor conductivity in the crystalline state, proteins were also non-conductive macromolecules. Then the BAS was added, the unconverted sites of antibodies were blocked, leading to a further decrease in peak current (curve d). In the presence of TeA, the peak current further decreases (curve e) because the antibodies bound to TeA to form immune complexes, resulting in steric hindrance and blocking electron transfer. From the CV characterization results (Table 1), it can be seen that the electrochemical immunosensor was successfully constructed.

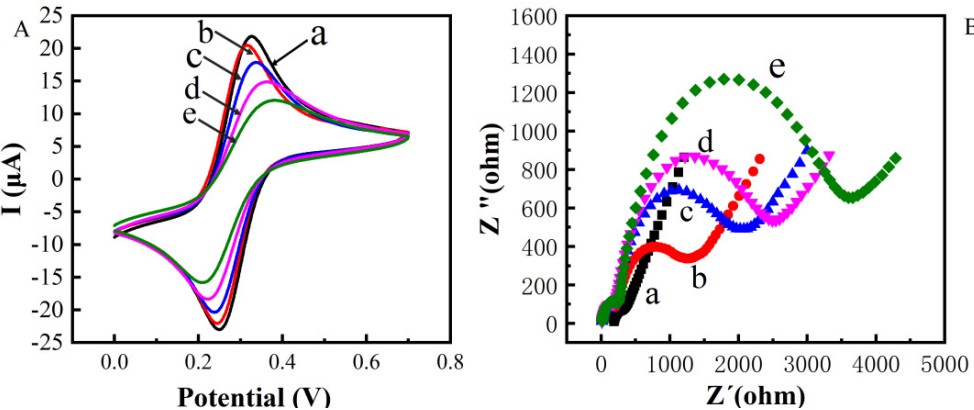

**Figure 3.** Cyclic voltametric characterization of constructing electrochemical immunosensor (**A**) Electrochemical impedance spectroscopy characterization of constructing electrochemical immunosensor (**B**) (a: bare Au; b: bare Au/ZnO; c: bare Au/ZnO/antibody; d: bare Au/ZnO/antibody/BSA; e: bare Au/ZnO/antibody/BSA/TeA).

Figure 3B shows the impedance (EIS) to further verify and characterize the fabrication process. The diameter of the semicircle in the high-frequency region of the Nyquist curve reflects the electron transfer impedance (Ret), which showed the electron transfer ability. The five curves (a), (b), (c), (d), and (e) in Figure 3B correspond to each layer of the

gold electrode surface modification operation. It could be seen that the modification was assembled on the electrode surface; the impedance increasing sequentially could also indicate the successful modification of each layer of material and the occurrence of immune responses.

**Table 1.** Comparison table of characterization results.

|  | $E_{Pa}$ (V) | $E_{Pc}$ (V) | $\Delta Ep$ (V) | $I_{Pa}$ (A) | $I_{Pc}$ (A) |
|---|---|---|---|---|---|
| bare Au | 0.25 | 0.33 | −0.08 | −23.13 | 21.84 |
| bare Au/ZnO | 0.25 | 0.32 | −0.07 | −21.84 | 20.36 |
| bare Au/ZnO/antibody | 0.23 | 0.33 | −0.10 | −20.45 | 17.96 |
| bare Au/ZnO/antibody/BSA | 0.22 | 0.35 | −0.13 | −18.33 | 15.02 |
| bare Au/ZnO/antibody/BSA/TeA | 0.21 | 0.37 | −0.16 | −15.84 | 11.89 |

### 3.4. Optimization of Experimental Conditions

The modification time of MBA on the electrode was related to the number of carboxyl groups introduced and the influence of immobilization of the ZnO-mAb on the electrode. It can be seen from Figure 4A that with the increase in modification time, the peak value of DPV current change gradually increased and stayed the same, indicating that the modification time of MBA reached equilibrium after 2 h. Therefore, the most suitable MBA modification time was selected as 2 h.

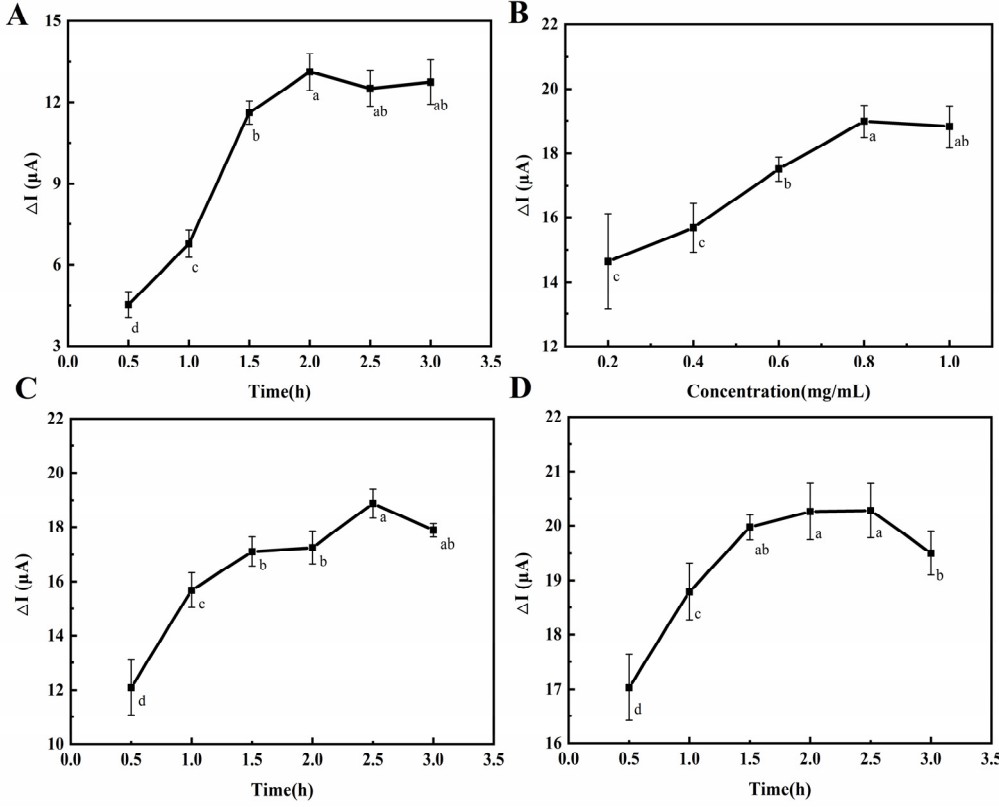

**Figure 4.** Effect of different modification times of MBA on differences in the peak value of the DPV current (**A**). Effect of ZnO-mAb concentrations on differences in the peak value of the DPV current (**B**). Effect of different modification times of ZnO-mAb on differences in the peak value of the DPV current (**C**). Effect of different modification times of BSA on differences in the peak value of the DPV current (**D**).

To obtain the best performance of the electrochemical immunosensor, the loading amount of ZnO-mAb on the electrode was investigated. It can be seen from Figure 4B that

with the increase in ZnO-mAb concentration, the peak value of DPV current change gradually increases. When the concentration was 0.8 mg/mL, it reached the equilibrium state, indicating that the electrode's loading of ZnO-mAb had reached saturation at this time.

It can be seen from Figure 4C that with the increase in the incubation time of ZnO-mAb, the peak value of DPV current change gradually increased and then reached equilibrium at 2.5 h. This may be related to the saturation of the electrode load. Competitive binding of the remaining unbound substances was related to the competitive binding, which will cause a part of the complex that was originally loaded on the electrode surface to be "squeezed", resulting in a reduction in the load on the electrode. Therefore, the incubation time of ZnO-mAb was 2.5 h.

After the electrode surface was saturated with ZnO-mAb, there were active sites, resulting in the non-specific recognition. Therefore, BSA was chosen as a blocking agent to occupy the remaining sites to prevent the occurrence of non-specific reactions. It can be seen from Figure 4D that the DPV current change increased within 1.5 h and then reached equilibrium, indicating that the remaining binding sites were all occupied. Therefore, the blocking time was 1.5 h. After 2.5 h, the DPV current change began to decrease, probably because the long blocking time caused itself to fall off the electrode surface.

### 3.5. The Sensitivity

Three concentration gradients of $10^{-4}$, $10^{-3}$, and $10^{-2}$ µg/mL were selected, and the precision of the method was investigated through repeated experiments. The relative standard deviation RSD ($n$ = 5) was between 3.05% and 6.31%, indicating that the method had a good precision. The constructed electrochemical sensor was used for the detection of TeA with a concentration range of $5 \times 10^{-6}$ µg/mL to 1µg/mL, and the results are shown in Figure 5A. The logarithmic value of TeA standard concentration (lgC) and the peak difference in DPV current ($\Delta I'$) had a good linear relationship, linear regression in Figure 5B. The equation was y = 1.35lgC + 25.67, $R^2$ = 0.99. A comparison of this method and other reported methods is shown in Table 2. The electrochemical immunosensor exhibited the lowest LOD of 0.01 ng/mL among the existing methods and had the range of 0.05 to 500 ng/mL, at least one order of magnitude wider than other methods.

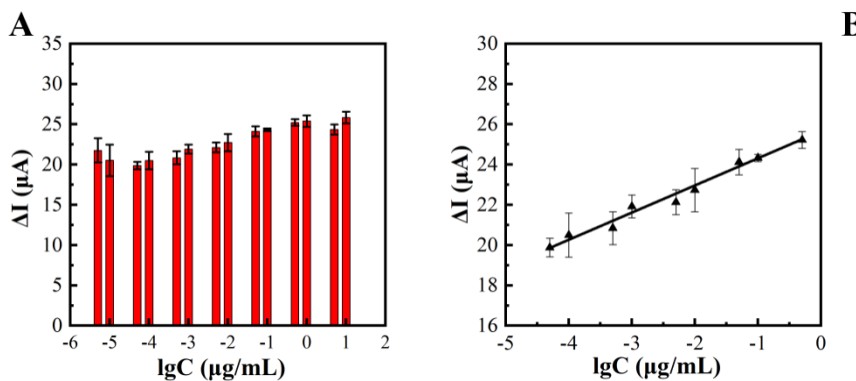

**Figure 5.** Relationship between TeA concentrations and differences in the peak value of the DPV current (**A**). Linear plot between Logarithm of TeA concentration versus current response (µA) (**B**).

**Table 2.** Comparison of the proposed aptasensor with those reported in the literature.

| Methods | LOD | Linear Range | |
|---|---|---|---|
| ELISA | 0.08 ng/mL | 0.26–25.90 ng/mL | [31] |
| ELISA | 0.39 ng/mL | | [6] |
| chemiluminescence | 0.2 ng/mL | 0.90–69.80 ng/mL | [32] |
| ELISA | 1 ng/mL | 3.56–96.24 ng/mL | [33] |
| fluorescence polarization | 0.13 μg/mL | 0.19–47.7 μg/mL | [34] |
| ELISA | 0.5 ng/mL | 1.70–36.40 ng/mL | [35] |
| electrochemical | 0.01 ng/mL | 0.05–500 ng/mL | This work |

### 3.6. The Specificity

Four kinds of Alternaria toxins similar in structure to TeA, Altenuene (ALT), Ten toxin (TEN), AME, AOH were used as interfering substances to investigate the specificity of this chemical immunosensor. As shown in Figure 6, the results showed that the same electrochemical response value was reached; the TeA concentration only needed to reach 15.23–25.07% of the interfering toxin concentration, indicating that this electrochemical immunosensor exhibited a good specificity.

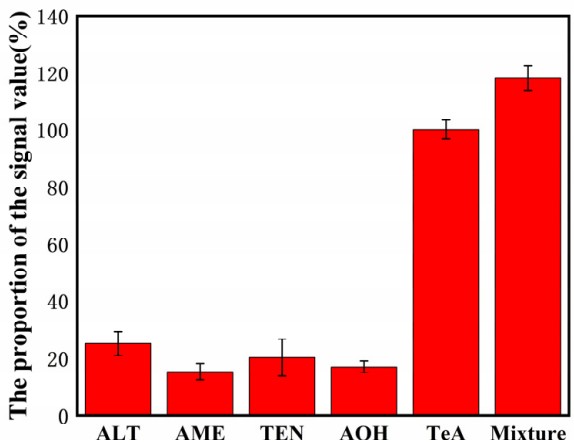

**Figure 6.** Selectivity of the biosensor detection of TeA (500 pg/mL) against the interference proteins: 50 ng/mL ALT, 50 ng/mL AME, 50 ng/mL TEN, and 50 ng/mL AOH.

### 3.7. Actual Sample Detection

Tomato and oranges were selected as the actual samples to investigate the application of this developed immunosensor. The detection results of the spiked samples are shown in Table 3. The average recovery of standard addition in tomatoes was 95.71–105.77%, and the average recovery of standard addition in citrus was 98.06–120.30%, indicating it showed good accuracy.

**Table 3.** Recovery test of TeA added to different samples.

| Sample | Added (μg/mL) | Recovery (%) | RSD (%) $n = 3$ |
|---|---|---|---|
| tomato | $10^{-2}$ | 104.23 | 6.30% |
| | $10^{-3}$ | 95.71 | 4.15% |
| | $10^{-4}$ | 105.77 | 7.94% |
| oranges | $10^{-2}$ | 98.06 | 8.13% |
| | $10^{-3}$ | 104.78 | 5.31% |
| | $10^{-4}$ | 120.30 | 8.67% |

## 4. Conclusions

In summary, we prepared modified ZnO and flower-shaped nano-ZnO to solve the high-loading problem of monoclonal antibodies. CV and EIS measurements confirmed the successful synthesis of Au/ZnO/antibody/BSA/TeA. Furthermore, the electrochemical immunosensor demonstrated excellent sensitivity with a linear range from $5 \times 10^{-5}$ μg/mL to $5 \times 10^{-1}$ μg/mL, and limit of detection down to $1.14 \times 10^{-5}$ μg/mL. At the same time, specificity studies showed that it had a particular anti-interference performance against toxins with a similar structure to TeA. A simple, cost-effective, and pollution-free nano-ZnO was successfully used to determine TeA with a low detection limit, which can pave the way for the construction of more efficient and sensitive sensors and biosensors toward detection of other analytes [36].

**Author Contributions:** Conceptualization, C.Z. and L.M.; methodology, W.L.; software, W.L.; formal analysis, C.Z. and C.D.; investigation, H.Z.; resources, X.L.; data curation, C.D.; writing—original draft preparation, C.Z.; writing—review and editing, Y.Z. (Ying Zhou) and T.G.; visualization, T.G.; supervision, L.M.; project administration, Y.Z. (Yuhao Zhang); funding acquisition, L.M. All authors have read and agreed to the published version of the manuscript.

**Funding:** This work was funded by the National Natural Science Foundation of China (grant number 32072137), Fundamental Research Funds for the Central Universities of China (grant number XDJK2020B044 and XDJK2020C052), Venture & Innovation Support Program for Chongqing Overseas Returnees grant number cx2018032).

**Institutional Review Board Statement:** Not applicable.

**Data Availability Statement:** The data presented in this study are available on request from the corresponding author.

**Conflicts of Interest:** The authors declare that they have no known competing financial interest or personal relationships that could have appeared to influence the work reported in this paper.

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
