# Peer review of "A High Sensitivity Electrochemical Immunosensor Based on Monoclonal Antibody Coupled Flower-Shaped Nano-ZnO for Detection of Tenuazonic Acid"

_agriculture, doi:10.3390/agriculture12020204_

Round 1
Reviewer 1 Report
A High Sensitivity Electrochemical Immunosensor Based on 2 Monoclonal Antibody Coupled Flower-shaped nano-ZnO for 3 Detection of Tenuazonic Acid
minor revision
Comments:
- In the introduction section, the authors emphasized the importance of ZnO and its importance in electrochemical sensor, also, some reported relevant articles are suggested to be mentioned, such as 10.1039/C9NJ02993A, 10.1039/C9TC07043E
- The paper is well written except for some minor grammar mistakes and typos which should be corrected before the final submission.
- The figures are not at standard. The overall quality of the figure should be improved. (Graphs have to be replotted for better clarity and check the abscissa – Fig.1C and Fig. 2)
- Figure captions are not proper and clear (too poor).
- Author should include the JCPDS card no in the XRD (Fig. 1C).
- Author should provide the comparison table for Section 3.3, which includes the EPa, EPc, ΔEp, IPa and IPc.
- The author should provide more literature report, especially electrochemical sensor (Tenuazonic Acid detection via electrochemically).
Reviewer 2 Report
The manuscript describes the development of an electrochemical immunosensor for tenuazonic acid. Overall, the written quality of the manuscript is below average. There are many truncated sentences and misspellings which need to be addressed by the authors. Therefore, I suggest authors to seek professional English revision. Moreover, there are many points of concern that need attention, namely:
- Abstract, line 21 and other instances throughout the text: tenuazonic acid abbreviature needs to be consistent all along the manuscript. Moreover, authors could also name thiosalicylic acid as 2-mercaptobenzoic acid, as the selected abbreviation was clearly based on this nomenclature;
- Introduction, lines 53 and 54: authors state: “More significantly, it could exhibit distinctive electrical, chemical, and optical properties, owing to relatively large band gap and exciton binding energy”. This statement is generic and offers little information for readers. I kindly ask authors to explain these alleged distinctive properties under the light of crafting an electrochemical immunosensor, such as the one whereupon the research is concerned;
- Introduction, lines 72 and 73: I kindly ask authors to explain why the development of an electrochemical immunosensor for tenuazonic acid using EDC/NHS protocol for surface chemistry and flower-like ZnO nanostructures opens up a “new research direction” in food and agricultural sciences. All these techniques are well understood and constantly applied in these fields, so authors should clearly remark the alleged advances of their work;
- Introduction (all section): the introduction is very brief (i.e., three paragraphs) and uninformative. Therefore, I suggest authors to expand it and comprehensively explore state-of-the-art outreaches regarding tenuazonic acid detection and the application of electrochemical immunosensors for this purpose;
- Methods, lines 84 and 85: it is customary to describe the methods in the past tense, thence I kindly ask authors to adequate it;
- Methods, lines 91 and 92, and several other instances throughout the manuscript: please correct all instances of misspelling, such as: “electrochemi-cal”, “electrochemicalworkstation”, “ascounter”, “workingelectrode” and others.
- Methods, lines 83, 104 and 105: please add the full name of TEOS and APTES, as well as BCA and other unidentified abbreviatures;
- Results and Discussion, lines 155 and 166: please check the formatting and correct the misspelling;
- Results and Discussion (overall): did authors also test EIS response as the analytic signal (i.e., increase in semicircle diameter/charge-transfer resistance)? This should be done and compared to the DPV responses. Moreover, in what solution was EIS performed? This information needs to be added;
- Results and Discussion (overall): authors separated results and discussion on different sections, however the discussion was partially presented on the results section, and on the discussion section there was a single paragraph summarizing the observations and providing little actual discussion of the results. Authors should comprehensively discuss their results and compare to recent literature.
- Conclusions (overall): authors did not add conclusions, even though their discussion was very brief and lacking in information. Therefore, I suggest authors all the aforementioned modifications and also to add a brief conclusion to their work.
Therefore, in view of the suggested changes, I suggest major revision.
Reviewer 3 Report
It is a good work, clear, well structured and with good electrochemical results. However discussion is poor and it is mandatory include any conclusions.
In my opinion the core of the paper must be clearly defined and it be resolved on the conclusions, that shall be supported by results. These topics are all of them on the text, however, could try to improve their linkages.
Biosensors is a trend topic on electrochemistry, Specificity and selectivity have to be alignment with sustainability and green chemistry. In this sense, authors would try introduce these approaches.
Immunosensor is not very original topic, however, it is a good work that can use in as reference in other paper about this matter.
I suggest some points on the document attached. I would try to focus the paper on analytical methods or real samples with the minimization of sample preparation.
In addition, the figures must be improved.

Round 2
Reviewer 2 Report
The authors made significant upgrades in the current version of the manuscript. Although the written quality could still be improved, I suggest acceptance owing to the small amount of required corrections.
